Identification of key pathways and genes underlying melatonin-enhanced drought tolerance in cotton

Zhong Xingyue
Han Aixia
Liusui Yunhao
Zhang Xin
Fu Wanwan
Wang Ziyu
Li Yuanxin
Cao Jing
Guo Yanjun coad25@foxmail.com
Zhang JingBo 18910445207@163.com
School of Life Sciences, Xinjiang Normal University , Ürümqi , Xinjiang Uygur Autonomous Region , China
Gelfand Mikhail
Electronic publication date: 2025 Sep 23
Publication date: 2025
Volume: 13
Electronic Location ID: e20005
Received 2025 May 21; Accepted 2025 Aug 7
Copyright: ©2025 Zhong et al.
Copyright year: 2025
Copyright holder: Zhong et al.
License: This is an open access article distributed under the terms of the Creative Commons Attribution License, which permits unrestricted use, distribution, reproduction and adaptation in any medium and for any purpose provided that it is properly attributed. For attribution, the original author(s), title, publication source (PeerJ) and either DOI or URL of the article must be cited.
License URL: https://creativecommons.org/licenses/by/4.0/

Keywords: Cotton, Drought, Melatonin, Physiological role, Molecular pathways

Funding: National Natural Science Foundation of China 32360489 Talent Development Fund of Xinjiang Uygur Autonomous Region (Tianchi Youth Talent Introduction Program) 2024 Annual Program for Youth Elite Talent Development at Xinjiang Normal University XJNUQB2024-08 This study was supported by the National Natural Science Foundation of China (Grant No. 32360489), the Talent Development Fund of Xinjiang Uygur Autonomous Region (Tianchi Youth Talent Introduction Program), and the 2024 Annual Program for Youth Elite Talent Development at Xinjiang Normal University (No. XJNUQB2024-08). The funders had no role in study design, data collection and analysis, decision to publish, or preparation of the manuscript.

==============================
Drought stress is a significant environmental constraint that adversely affects the growth of upland cotton (Gossypium hirsutum) by inducing complex physiological disruptions. Emerging research evidence indicates that melatonin (MT), as a plant growth regulator, is extensively involved in the process of plant stress resistance regulation. This study explored the role of exogenous MT in enhancing drought tolerance in cotton, employing both physiological parameter analysis and transcriptomic profiling to unravel the underlying mechanisms of its stress mitigation effects. The results demonstrated that MT treatment significantly enhanced drought resistance in cotton plants by upregulating leaf superoxide dismutase (SOD), peroxidase (POD), and catalase (CAT) activities, elevating proline (PRO) content, and decreasing malondialdehyde (MDA) accumulation, thus confirming its physiological role in alleviating drought stress. Transcriptome analysis revealed that MT specifically modulates the “plant circadian rhythm”, “thiamine metabolism”, and “taurine and hypotaurine metabolism” pathways under drought stress conditions, thereby playing a pivotal role in drought adaptation. Further analysis of the 276 differentially expressed genes (DEGs) specifically modulated by MT under drought stress, combined with co-expression network analysis, identified two MT-specific induced Basic Helix-Loop-Helix (bHLH) family transcription factors (GhPIF8 and GhMYC5, gene IDs: Ghi_A11G05431 and Ghi_D03G05926) as key regulatory candidates in MT-mediated drought tolerance. This study establishes a theoretical framework for understanding the physiological and molecular mechanisms underlying MT-mediated drought tolerance in cotton, while also informing practical applications of MT in cotton agriculture.

Introduction

Cotton (Gossypium hirsutum), a dual-purpose economic crop valued for both fiber and oil production (Hu et al., 2025), holds strategic importance in the global agricultural sector. China’s Xinjiang region has emerged as the country’s core cotton production base by leveraging its unique solar-heat advantages and large-scale cultivation capabilities, contributing approximately 92% of the nation’s total cotton output and providing stable raw material support for the textile industry. In recent years, continuous advancements in agricultural technology have significantly improved cotton yield per unit area. Nevertheless, the interplay between the region’s unique geospatial attributes and global climate variability has elevated drought stress to a pivotal constraint impeding yield improvement in local cotton production (Zhou et al., 2025a). Drought stress perturbs the hydrological homeostasis and decreases their water-use efficiency in cotton plants, triggering a cascade of physiological and biochemical responses, including membrane lipid peroxidation and osmotic regulation imbalance. Short-term extreme drought or prolonged water deficit can lead to significant yield reduction or even plant mortality. Research evidence suggests that cotton plants predominantly employ integrated regulatory processes, including stomatal modulation, activation of antioxidant defense cascades, accumulation of osmoprotective compounds, and hormonal signal transduction pathways, to counteract drought stress (Li et al., 2025a). To address these challenges, the application of novel exogenous plant drought-resistant regulators to synergistically enhance endogenous stress tolerance has emerged as a promising strategy to stabilize cotton production (Cen et al., 2020; Abdullah et al., 2025).

Melatonin (N-acetyl-5-methoxytryptamine, MT), a naturally occurring indoleamine, is found throughout eukaryotic organisms (Reiter et al., 2025). Discovered in plants in 1995 (Ye et al., 2025), its multifaceted roles have since been extensively characterized. MT serves as a regulator of diverse developmental processes—encompassing seed germination, root morphogenesis, photosynthetic efficiency, leaf senescence, floral transition, and fruit maturation (Mansouri, Koushesh Saba & Sarikhani, 2023; Yu et al., 2024; Kang et al., 2025; Zhang et al., 2025b; Lv et al., 2025)—and also functions as a pivotal determinant of plant immunity. For instance, Li et al. (2022a) showed that MT suppresses bacterial angular leaf spot (BALS) in cucumber by inhibiting the proliferation of Pseudomonas syringae pv. lachrymans (Psl), activating disease resistance signaling pathways, enhancing the antioxidant defense system, inducing the expression of stress-responsive genes, and balancing growth-defense trade-offs. When combating apple blotch (Diplocarpon mali), MT treatment enhances host resistance through the regulation of dynamic hydrogen peroxide (H2O2) homeostasis and activation of defense enzymes (Yin et al., 2013). Moreover, mechanistic investigations in cotton reveal that MT transcriptionally co-regulates key genes within the phenylpropanoid biosynthesis, mevalonate (MVA), and gossypol biosynthetic pathways. This coordinated regulation collectively stimulates lignin deposition and gossypol accumulation, thereby conferring enhanced resistance against Verticillium dahliae infection (Li et al., 2019a).

MT has markedly improved plants’ resilience to diverse abiotic stresses, such as cold, heat, drought, salinity-alkalinity, and heavy metals (Kour et al., 2024; Tian et al., 2025; Asghari, Hoseinzadeh & Mafakheri, 2025; Zhou et al., 2025b). MT enhances drought tolerance in maize seedlings by stabilizing chloroplast structure, sustaining cell expansion, protecting the cell wall, improving stomatal traits, and mitigating the negative effects of reactive oxygen species (ROS) (Muhammad et al., 2023). In research involving Camellia hainanica, MT treatment significantly increased the plants’ tolerance to drought stress simulated with polyethylene glycol, achieved by elevating antioxidant enzyme activities and accumulating secondary metabolites (Ge et al., 2025). In applied research on wheat, a coating treatment with 100 µM MT successfully activated the antioxidant enzyme system, thereby increasing yield during drought stress (Li et al., 2025b). Zhang, Liang & Dong (2024) discovered that MT enhances cotton’s waterlogging tolerance and reduces yield losses caused by waterlogging stress through a series of physiological, biochemical, and molecular changes involving quiescence adaptation and compensatory growth strategies.

Transcriptomic analyses suggest that exogenous MT confers salt tolerance to banana by orchestrating the regulation of multiple metabolic pathways, including amino sugar/nucleotide sugar interconversion, phenylalanine metabolism, cyanoamino acid metabolism, starch-sucrose partitioning, and linoleic acid metabolism (Wei et al., 2022a). MT can alleviate drought stress by activating the tryptophan metabolism and flavonoid biosynthesis pathways in wheat (Li et al., 2024), as well as synergistically promoting the biosynthesis of jasmonic acid and lignin (Luo et al., 2023b). In maize, MT upregulates genes involved in flavonoid synthesis in the roots, thereby stimulating root system development and enhancing drought adaptation (Wang et al., 2023a). Research findings indicate that MT confers drought tolerance to naked oat (Avena nuda) through the targeted regulation of drought-responsive transcription factor expression and genes associated with abscisic acid (ABA) signaling pathways (Zhang et al., 2022). In contrast, MT alleviates drought-induced damage in tomato (Solanum lycopersicum) through promoting sucrose biosynthesis and suppressing ABA biosynthesis (Jahan et al., 2024). Investigations into the molecular mechanisms reveal that PtbHLH28 and PtABF4 influence plant drought tolerance by regulating MT biosynthesis by PtCOMT5 and root development mediated (Zhu et al., 2025). Furthermore, the CGA1-SNAT regulatory module in wheat appears to participate in the drought resistance response via the cytokinin-mediated MT synthesis pathway (Shamloo-Dashtpagerdi, Lindlöf & Aliakbari, 2025). Exogenous MT treatment enhances freezing tolerance by inducing the expression of the C2H2-type zinc finger transcription factor AtZAT6 (Shi & Chan, 2014). Additionally, MT promotes cell wall remodeling and adventitious root development by activating the transcription factor SlSBRL1 in tomato, while SlSBRL1 amplifies MT signaling through positive feedback regulation by further upregulating its own expression (Liu et al., 2025a). Li et al. (2025c) revealed that MT demonstrated that MT inhibits ethylene biosynthesis and delays fruit ripening through dual mechanisms: downregulating MdREM10 expression to indirectly inhibit the transcription of the key ethylene biosynthesis gene MdACS1, and directly repressing MdACO1 transcription via decreased MdZF32 expression. These findings collectively demonstrate that MT exerts critical roles in plant abiotic stress adaptation and other biological processes by orchestrating multiple metabolic pathways and modulating key gene expression.

Previous studies have indicated that exogenous MT plays a role in enhancing drought resistance in cotton. Specifically, exogenous MT alleviates drought-induced damage and improves yield by enhancing the function of antioxidant enzyme systems, increasing photosynthetic efficiency, regulating ABA synthesis, and optimizing osmotic adjustment and root development (Hu et al., 2021; Yang et al., 2023; Zhu et al., 2023; Zhu et al., 2024b; Zhang et al., 2025a). Additionally, it modulates cotton drought resistance through regulating methylglyoxal (MGO) homeostasis and autophagic activity in seeds, maintaining anther carbohydrate balance, and enhancing the activities of enzymes related to sugar and nitrogen metabolism during the boll stage (Hu et al., 2020; Supriya et al., 2022; Khattak et al., 2023; Dake et al., 2025). However, the effects of exogenous MT on drought resistance in cotton seedlings and its underlying molecular mechanisms have not yet been reported, and further in-depth investigation is warranted.

In the present study, we investigated the role of exogenous MT in improving seedling-stage drought resistance in cotton through integrated phenotypic and physiological analyses, focusing on antioxidant enzyme activities and osmoregulatory substance accumulation. Transcriptomic sequencing was employed to identify the key metabolic pathways regulated by MT in cotton under drought conditions. Furthermore, weighted gene co-expression network analysis (WGCNA) revealed potential candidate effector genes involved in MT-mediated drought response. These findings advance our understanding of the molecular mechanisms underlying MT-regulated drought tolerance in cotton and provide a theoretical foundation for its application in cotton cultivation under water-limited environments.

Materials & Methods

Plant materials and growth conditions

The cotton cultivar used in this study was Zhongmian 113, which is a major promoted cultivar in the Xinjiang region. Cotton seeds were purchased from Zhongmian Group Aksu Agricultural Development Co., Ltd. Cotton seeds were subjected to an aseptic germination protocol involving sequential disinfection with 75% (v/v) ethanol (5 min) and 1% (v/v) sodium hypochlorite (30 min), interspersed with three sterile distilled water washes. Post-disinfection, seeds were dark-phase stratified between moistened filter membranes at 28 °C for 48 h. Germinated propagules were aseptically transferred to pre-calibrated pots containing a 3:1:1 (v/v/v) matrix of peat-based growth medium, loam, and phyllosilicate substrate. Cultivation occurred under controlled phytotron conditions (25/22 °C diurnal cycle; 16/8 h photoperiod).

Experimental design

The experiment commenced when the cotton plants reached the three-leaf and one-apical-bud stage, at which point watering was withheld. Prior investigations have shown that 100 µM MT represents the most effective concentration for counteracting the growth—inhibitory effects of drought on cotton seedlings (Bai et al., 2020; Khan et al., 2024). Signals generated by foliar application of MT can be transmitted among plant organs (Huang et al., 2019) and this treatment effectively alleviates the inhibitory effects of drought stress on plant growth (Ahmad et al., 2019). Based on these findings, we selected foliar spraying of 100 µM MT as the experimental treatment. MT was obtained from Solarbio Company in Beijing, China. MT was applied uniformly to cotton leaves using a same sprayer. Each plant was sprayed four times in total, with each leaf receiving one application, ensuring that the leaves were thoroughly wetted without runoff to guarantee uniform adherence of MT. To investigate the pathways and genes specifically regulated by MT under drought stress, we designed the following experiments based on the methodology described by Duan et al. (2022):

(i). Control group (CK): Seedlings maintained under standard irrigation protocols with foliar application of water.

(ii). MT-treated control group (CK_MT): Standard irrigation maintained alongside 100 µM MT phyllosphere supplementation.

(iii). Drought stress group (DS): Watering was ceased, and seedlings were sprayed with distilled water on the leaf surface.

(iv). MT-treated drought stress group (DS_MT): Water withholding regime combined with 100 µM MT phyllosphere delivery.

Foliar spray treatments were administered every 48 h on days 1, 3, and 5 after water withholding. Since plants had suffered drought stress by day 6 after water withholding, the spray frequency was adjusted to once every 24 h from days 6 to 8 to avoid potential MT wash-off during drought (Li et al., 2022c; Khan et al., 2024). Absolute soil water content (ASWC) was used to evaluate the drought stress experienced by cotton plants. Nine pots of plants were randomly selected from each group and weighed, with the weight recorded as WW1. The ASWC (%) was calculated using the formula: (WW1 − (dry weight of the mixed nutrient soil + pot weight))/dry weight of the mixed nutrient soil) ×100%. When the ASWC of the drought treatment group dropped below 10%, cotton leaf samples were collected from all groups.

Relative water content of leaves under drought stress

Leaves were collected from cotton plants in each treatment group, with three biological replicates per treatment. The formula for calculating leaf relative water content (RWC) is: ((WF−WD)/(WS−WD)) × 100%. The specific measurement steps are as follows: (i) Excise fresh leaf tissue and weigh it immediately to obtain the fresh weight (WF); (ii) Immerse the leaves in deionized water and incubate in the dark for 12 h to allow full hydration, then determine the turgid weight (WS); (iii) Dry the leaves in an oven at 65 °C until a constant weight is achieved to obtain the dry weight (WD) (Ahmad et al., 2019).

Measurement of SOD activity

Leaves were collected from cotton plants in each treatment group, with three biological replicates per treatment. Superoxide dismutase (SOD) activity was determined using a visible spectrophotometric method following the manufacturer’s instructions of the SOD Assay Kit (Grace, Suzhou, China). Briefly, SOD activity was calculated by measuring the formazan dye content at a wavelength of 450 nm (λ = 450 nm).

Measurement of POD activity

Leaves were collected from cotton plants in each treatment group, with three biological replicates per treatment. Peroxidase (POD) activity was determined using a visible spectrophotometric method following the manufacturer’s instructions of the POD Assay Kit (Grace, Suzhou, China). Briefly, at a wavelength of 470 nm, POD activity was calculated by measuring the change in absorbance during the process where POD catalyzes the oxidation of guaiacol by H2O2 to form a reddish-brown product.

Measurement of CAT activity

Leaves were collected from cotton plants in each treatment group, with three biological replicates per treatment. Catalase (CAT) activity was determined using a visible spectrophotometric method following the manufacturer’s instructions of the CAT Assay Kit (Grace, Suzhou, China). Briefly, CAT activity was spectrophotometrically quantified via H2O2 dismutation kinetics at λ510 nm.

Measurement of PRO content

Leaves were collected from cotton plants in each treatment group, with three biological replicates per treatment. Proline (PRO) content was determined using a visible spectrophotometric method following the manufacturer’s instructions of the PRO Assay Kit (Grace, Suzhou, China).

Measurement of MDA content

Leaves were collected from cotton plants in each treatment group, with three biological replicates per treatment. Malondialdehyde (MDA) content was determined using a visible spectrophotometric method following the manufacturer’s instructions of the MDA Assay Kit (Grace, Suzhou, China).

RNA extraction and sequencing

RNA samples were extracted from cotton leaves subjected to various treatments, with three biological replicates representing each condition. RNA isolation was performed using a plant RNA extraction kit (FOREGENE, Chengdu, China), followed by reverse transcription into cDNA using a reverse transcription kit (FOREGENE, Chengdu, China). RNA integrity verification was executed through tripartite analytical verification: NanoDrop™ microvolume UV-Vis spectrophotometry (PEQLAB Biotechnologie GmbH, Erlangen, Germany), Qubit® 2.0 Fluorometric Quantification System (Thermo Fisher Scientific, Waltham, MA, USA), and capillary electrophoresis-based macromolecular profiling (Agilent 2100 Bioanalyzer™, Agilent Technologies, Santa Clara, CA, USA). The cDNA fragments underwent magnetic bead-based purification with size fractionation (200 ± 20 bp) employing AMPure XP paramagnetic particles. Post-amplification processing yielded sequencing-ready libraries that were subjected to high-throughput sequencing via the DNBSEQ-T7 platform (MGI Tech, Shenzhen, China) under 2 ×150 bp paired-end configuration. Primary signal conversion was demultiplexed through the Illumina CASAVA pipeline (v1.8.2), with raw sequencing data archived in FASTQ-compliant binary format.

Raw sequencing data underwent quality control preprocessing through fastp (v0.23.4), implementing: (i) adapter contamination excision, (ii) filtration of ambiguous nucleotide calls (N), and (iii) elimination of suboptimal-quality sequences (Phred score < 30). Post-processing quality metrics (Q20, Q30) and nucleotide composition (GC content) were computationally assessed via fastp (v2.2.1), with quality-filtered sequencing reads advanced to downstream bioinformatic workflows.

Transcriptomic reconstruction was executed via StringTie (v2.2.1), with RSEM (v1.3.3) implementing length-normalized FPKM quantification to profile transcript abundance.

The statistical power for transcriptome samples was estimated using the RNA-Seq Power Analysis Tool (https://rodrigo-arcoverde.shinyapps.io/rnaseq_power_calc/), with a calculated power of 83.6%.

Intergroup differential expression profiling across four experimental cohorts was executed through DESeq2 (v1.22.1) implementation, employing stringent statistical thresholds (|log2FC| ≥1; FDR-adjusted p < 0.05) to delineate significant differentially expressed genes (DEGs). Functional annotation enrichment (Gene Ontology) was conducted via Goatools v1.4.4, with parallel Kyoto Encyclopedia of Genes and Genomes (KEGG) orthology pathway mapping implemented through custom R-based algorithms. Systemic co-expression network modeling was subsequently applied using WGCNA v1.63 within a multi-omics integration framework.

Quantitative real-time PCR analysis

Ten DEGs specifically induced by MT under drought stress were selected. Using gene-specific primers listed in Table S1, RNA sequencing results were validated by reverse transcription quantitative polymerase chain reaction (RT-qPCR). The GhHIS3 gene was used as the reference gene, and relative expression levels were calculated using the 2−ΔΔCT method.

Statistical analysis

Microsoft Excel 2021 was used for data collation and analysis. Statistical evaluation was implemented in GraphPad Prism 9 (v9.5.1) employing one-way ANOVA with Tukey post hoc tests  to determine intertreatment variability in physiological indicator parameters. Additionally, bar charts were generated using GraphPad Prism 9 software to visually present the results.

Results

Effects of MT treatment on growth and physiological parameters of cotton seedlings under drought stress

To elucidate the regulatory mechanisms of exogenous MT priming on drought resilience in cotton, a comparative physiology trial was conducted where MT-treated versus non-treated cohorts underwent controlled hydric deficit imposition. Phenotyping revealed isomorphic morphological conformity between DS (drought-stressed control plants) and DS_MT (drought-stressed plants with MT treatment) during pre-stress acclimation phases.

Figure 1 Phyllo-physiological modulation by exogenous MT supplementation in cotton seedlings under progressive hydric deficit.

(A) Phenotype, (B) RWC, (C) PRO content, (D) SOD activity, (E) POD activity, and (F) MDA content. CK is the control group, CK_MT is the group treated with 100 µM exogenous MT under normal conditions, DS is the group exposed to drought stress, and DS_MT is the group treated with 100 µM exogenous MT under drought stress. The bar charts show the standard deviation (SD) ±mean values from repeated measurements. Statistical significance was determined through Student’s t-test, employing a tiered significance threshold system (*p < 0.05; **p < 0.01; ***p < 0.001).

However, following 11-day progressive drought imposition, the DS group manifested pronounced leaf wilting and substantial growth suppression, while the DS_MT group maintained better phenotypic integrity, indicating that exogenous MT application alleviated drought-induced adverse effects (Fig. 1A). Following experimental interventions, foliar physiological indices were quantified across non-supplemented and MT-supplemented cohorts undergoing progressive hydric deficit. Compared to CK (well-watered control plants) and CK_MT (well-watered plants with MT treatment), drought stress significantly reduced the RWC in both DS and DS_MT seedlings. Nevertheless, MT application mitigated this reduction (Fig. 1B). Furthermore, the DS group displayed significantly lower PRO content, POD, and CAT activities compared to the DS_MT group (Figs. 1C, 1E, 1F). Although SOD activity in the DS group was lower than that in the DS_MT group, the difference was not significant (Fig. 1D). Moreover, MDA levels were markedly higher in DS than in DS_MT (Fig. 1G). These findings demonstrate that MT treatment enhances antioxidant capacity and osmotic regulation in cotton plants while alleviating drought-triggered cellular damage, thereby improving drought tolerance.

Transcriptomic profiling of MT-treated cotton seedlings under well-watered and drought stress conditions

First, we extracted leaf RNA from the CK, CK_MT, DS, and DS_MT groups, and subsequently evaluated sample integrity by analyzing the RNA quality number (RQN) of each group’s RNA. The results showed that the RQN of all samples was ≥7.0, indicating that the RNA quality met the experimental requirements (Table S2). To delineate the molecular regulatory networks mediating MT-primed drought adaptation in cotton, foliar RNA isolates from the above four groups underwent high-throughput transcriptomic profiling. High-quality sequencing libraries were successfully prepared, with individual cDNA libraries generating 40.35–44.41 million raw sequencing reads. Post-filtering processes involving adapter trimming, ambiguous base removal, and quality thresholding retained 76.90 Gb of refined sequencing data, achieving >6.00 Gb/sample. Nucleotide composition analysis revealed GC proportions spanning 43.52–44.44%, with sequence quality metrics (Q20/Q30) surpassing empirical thresholds of 98.42% and 94.89%, respectively (Table S3).

Principal component analysis (PCA) of transcriptomic data across treatment groups revealed distinct clustering, indicating significant transcriptional divergence among the four experimental conditions (Fig. S1A). Furthermore, hierarchical clustering analysis of gene expression profiles demonstrated clear separation between treatment groups, corroborating the PCA results (Fig. S1B). Collectively, these high-quality transcriptomic datasets provide a robust foundation for subsequent mechanistic analyses.

Analysis of gene transcription level changes induced by MT under normal watering and drought stress conditions

To comprehensively investigate the changes in gene expression induced by MT, we identified DEGs across the various comparison groups. The identification criteria were established at |logFC| ≥ 1 and a p-value <0.05. In the comparisons between the CK and the DS, the CK and the CK_MT, and the DS and the DS_MT, we identified 13,567 DEGs (of which 5,496 were upregulated and 8,071 were downregulated), 3,076 DEGs (with 1,725 upregulated and 1,351 downregulated), and 3,145 DEGs (including 1,808 upregulated and 1,337 downregulated), respectively (Figs. 2A–2C).

Figure 2 Volcano plots of DEGs in cotton leaves subjected to drought and MT treatments.

(A) CK vs. DS, (B) CK vs. CK_MT, (C) DS vs. DS_MT, (D) CK_MT vs. DS_MT. CK is the control group, CK_MT is the group treated with 100 µM exogenous MT under normal conditions, DS is the group exposed to drought stress, and DS_MT is the group treated with 100 µM exogenous MT under drought stress. The abscissa delineates log2-transformed fold change (FC) in transcript abundance between comparative cohorts, with the ordinate quantifying Benjamini–Hochberg corrected p-values (−log10 transformed). Chromatic demarcation identifies upregulated transcripts (crimson nodes), downregulated transcripts (zure nodes), and non-significant expression variance (neutral nodes).

Validation of the accuracy of RNA-seq data via qRT-PCR

To verify the accuracy of DEGs identified by RNA-seq, 10 differentially DEGs induced by MT under drought stress (Ghi_A11G05431, Ghi_D03G05926, Ghi_D03G02356, Ghi_A01G09861, Ghi_A01G09866, Ghi_A08G01351, Ghi_D03G07201, Ghi_D06G09006, Ghi_D05G05851, Ghi_A06G09411) were selected for validation via qRT-PCR (Table S4, Fig. S2). The expression trends identified in the transcriptome data were generally consistent with the results of qRT-PCR analysis, indicating that the transcriptome data have high reliability.

Expression analysis of genes encoding SOD, POD, and CAT

Physiological assays showed elevated SOD, POD, and CAT activities in MT-treated cotton under drought. To explore whether MT also modulates their transcriptional levels, we analyzed the gene expression profiles of these enzymes in the transcriptome data. The analysis revealed that, compared to the DS group, the DS_MT group showed significant upregulation of multiple genes encoding SOD, POD, and CAT. Notably, the number of upregulated POD genes was the highest among these three gene families (Fig. S3). To further investigate whether the transcriptional changes of SOD, POD, and CAT correlate with their enzymatic activity variations, we analyzed the Pearson correlation between the expression levels of the top three most significantly upregulated genes in each gene family (from the DS_MT group compared to the DS group) and their corresponding enzyme activities. The results showed that the Pearson correlation coefficients were 0.85, 0.99, and 0.91 for SOD, POD, and CAT, respectively. The expression trends identified in the transcriptome data were generally consistent with the physiological index analysis results, confirming that MT also modulates the transcriptional levels of SOD, POD, and CAT-encoding genes.

GO enrichment analysis of DEGs induced by MT treatment under normal watering and drought stress

Functional annotation of MT-responsive DEGs was performed through GO term enrichment profiling to decode MT-mediated transcriptional reprogramming. The DEGs were categorized according to GO terms, which were classified into three primary categories: cellular component (CC), molecular function (MF), and biological process (BP).

In the comparison of CK vs. DS, a total of 83 GO terms were enriched within the CC category, predominantly focusing on cellular anatomical entities, membranes, plasma membranes, extracellular regions, plastids, chloroplasts, supramolecular complexes, and plastid membranes (Fig. 3A). Under the MF category, 384 GO terms were identified, with significant enrichment in molecular_functions, catalytic activities, ion binding, small molecule binding, anion binding, nucleoside phosphate binding, and nucleotide binding (Fig. 4A). The BP category revealed 550 enriched entries, primarily associated with biological_processes, metabolic activities, biological regulation, regulation of cellular processes, biosynthetic processes, organic substance biosynthesis, phosphorus metabolism, and phosphate-containing compound metabolism (Fig. 5A).

Figure 3 GO enrichment analysis of DEGs classified into CC.

(A) CK vs. DS (B). CK vs. CK_MT (C). DS vs. DS_MT (Enriched terms were filtered using a Benjamini–Hochberg corrected p-value threshold (FDR < 0.05). The ordinate denotes pathway nomenclature, with the abscissa quantifying enriched DEG counts. Chromatic intensity scales proportionally to the -log10-transformed p-value magnitudes.).

Figure 4 GO enrichment analysis of DEGs classified into MF.

(A) CK vs. DS (B) CK vs. CK_MT (C) DS vs. DS_MT (Enriched terms were filtered using a Benjamini–Hochberg corrected p-value threshold (FDR < 0.05). The ordinate denotes pathway nomenclature, with the abscissa quantifying enriched DEG counts. Chromatic intensity scales proportionally to the -log10-transformed p-value magnitudes.).

Figure 5 GO enrichment analysis of DEGs classified into BP.

(A) CK vs. DS (B) CK vs. CK_MT (C) DS vs. DS_MT (Enriched terms were filtered using a Benjamini–Hochberg corrected p-value threshold (FDR < 0.05). The ordinate denotes pathway nomenclature, with the abscissa quantifying enriched DEG counts. Chromatic intensity scales proportionally to the -log10-transformed p-value magnitudes.)

In the CK vs. CK_MT comparison, 13 GO terms were enriched in the CC category, primarily related to membranes, plasma membranes, extracellular regions, apoplasts, membrane sides, extracellular spaces, peroxisomes, and external encapsulating structures (Fig. 3B). The MF category showed the enrichment of 181 GO terms, mainly concentrated on molecular_functions, catalytic activities, ion binding, transferase activities, hydrolase activities, small molecule binding, anion binding, and nucleoside phosphate binding (Fig. 4B). In the BP category, 250 GO terms were enriched, focusing on phosphorus metabolism, phosphate-containing compound metabolism, phosphorylation, lipid metabolism, carbohydrate metabolism, catabolic processes, organic substance catabolism, and cellular lipid metabolism (Fig. 5B).

In the analysis of DS vs. DS_MT, 41 GO terms were enriched in the CC category, predominantly involving membranes, plasma membranes, organelle membranes, bounding membranes of organelles, cytosol, extracellular regions, plastid membranes, and outer membranes (Fig. 3C). The MF category revealed 258 enriched GO terms, emphasizing molecular functions, catalytic activities, oxidoreductase activities, transporter activities, transmembrane transporter activities, DNA-binding transcription factor activities, tetrapyrrole binding, and glycosyltransferase activities (Fig. 4C). The BP category showed 351 enriched GO terms, largely associated with metabolic processes, responses to stimuli, small molecule metabolism, carbohydrate metabolism, catabolic processes, organic substance catabolism, lipid metabolism, and oxoacid metabolism (Fig. 5C).

Analysis of these three datasets indicates that, within the BP category, the DEGs in the DS vs. DS_MT comparison were particularly enriched in organic acid metabolic processes, carboxylic acid metabolic processes, small molecule biosynthetic processes, responses to abiotic stimuli, and amino acid metabolic processes. This suggests that MT may enhance the drought resistance of cotton by modulating the expression of genes associated with these biological processes.

KEGG enrichment analysis reveals MT-Induced DEGs

To decode pivotal signaling axes through which MT mediates drought resilience, KEGG orthology mapping was conducted on DEGs meeting stringent significance criteria (FDR-adjusted P < 0.05). In CK vs. DS, DEGs were enriched in 64 KEGG pathways, primarily plant hormone signal transduction, the mitogen—activated protein kinase (MAPK) signaling pathway—plant, biosynthesis of cofactors, starch and sucrose metabolism, motor proteins, amino sugar and nucleotide sugar metabolism, glycolysis/gluconeogenesis, pyruvate metabolism (Fig. 6A).

Figure 6 The KEGG enrichment analyses of DEGs.

(A) CK vs. DS (B) CK vs. CK_MT (C) DS vs. DS_MT. The ordinate denotes KEGG pathway nomenclature, with the abscissa quantifying normalized enrichment ratios. Node diameters correlate with enriched DEG cardinality, while chromatic gradients scale with Benjamini–Hochberg adjusted p-value magnitudes (−log10 transformed).

For CK vs. CK_MT, DEGs were enriched in 34 KEGG pathways, including plant hormone signal transduction, cutin, suberine and wax biosynthesis, MAPK signaling pathway—plant, phenylpropanoid biosynthesis, plant-pathogen interaction, pentose and glucuronate interconversions, starch and sucrose metabolism, alpha-Linolenic acid metabolism (Fig. 6B).

In DS vs. DS_MT, DEGs were enriched in 44 KEGG pathways, predominantly MAPK signaling pathway—plant, starch and sucrose metabolism, glyoxylate and dicarboxylate metabolism, glycolysis / gluconeogenesis, amino sugar and nucleotide sugar metabolism, pyruvate metabolism, phenylpropanoid biosynthesis, flavonoid biosynthesis. Analysis of the three datasets revealed that the DEGs from DS vs. DS_MT were specifically enriched in pathways related to Circadian rhythm—plant, Thiamine metabolism, Taurine and hypotaurine metabolism (Fig. 6C).

Figure 7 Expression analysis of genes involved in thiamine metabolism and taurine/hypotaurine metabolism.

(A) Thiamine metabolism (B) Taurine and hypotaurine metabolism. Chromogenic matrix elements (red) demarcate MT-induced transcriptional upregulation, whereas azure elements denote pharmacological downregulation. Heatmap intensity gradients correlate with log2FC magnitudes, quantifying MT-mediated differential expression patterns.

Comparative analysis of enriched pathways across all three groups revealed that only the DEGs in the DS vs. DS_MT group were enriched in the pathways of circadian rhythm—plant, thiamine metabolism (ghi00730), and taurine and hypotaurine metabolism (ghi00430) (Fig. 6). Notably, a total of 16 genes were significantly enriched in the “Circadian rhythm—plant” pathway, with 10 genes being significantly downregulated, including four genes encoding CHS (gene IDs: Ghi_A02G00546, Ghi_D10G07831, Ghi_D10G07826, Ghi_D05G09351), two genes encoding COL (gene IDs: Ghi_A13G13631, Ghi_D03G06171), two genes encoding PIF8 (gene IDs: Ghi_A07G03966, Ghi_D07G03841), one gene encoding CRY (gene ID: Ghi_D05G08691), and one gene encoding FT (gene ID: Ghi_A08G14081). Additionally, six genes were significantly upregulated, namely four genes encoding LHY1 (gene IDs: Ghi_D12G08106, Ghi_A11G05556, Ghi_D12G08111, Ghi_D11G05836) and two genes encoding HY5 (gene IDs: Ghi_A08G14546, Ghi_D08G13746). In the thiamine metabolism pathway, nine upregulated genes were significantly enriched, including five genes encoding Thi1 (Gene IDs: Ghi_D06G09006, Ghi_D05G05851, Ghi_A01G03361, Ghi_A05G10696, Ghi_A06G09411), two genes encoding ThiC (gene IDs: Ghi_A03G07201, Ghi_A03G03086), and two genes encoding DXS (gene IDs: Ghi_A08G01351, Ghi_T08G01366). These enzymes are crucial for the biosynthesis of thiamine phosphate (Fig. 7A). In the taurine and hypotaurine metabolism pathway, five genes were significantly enriched: three downregulated genes encoding flavin-containing monooxygenase 1 (FMO1) (Gene IDs: Ghi_A01G09861, Ghi_A01G09866, Ghi_D01G09256) are involved in the process of taurine biosynthesis via hypotaurine oxygenation; the other two upregulated genes encoding γ-glutamyl transpeptidase (GGT) (Gene IDs: Ghi_D03G02356, Ghi_A06G03736) participate in the process of glutamyl group transfer to taurine (Fig. 7B). These MT-specific metabolic inductions under drought suggest that MT confers drought tolerance to cotton by modulating thiamine metabolism, as well as taurine and hypotaurine metabolism.

WGCNA analysis identified key genes mediated by MT that are associated with the response to drought stress

To identify key effector genes underlying MT-enhanced drought tolerance, we performed WGCNA on DEGs across all comparison groups. Four co-expression modules (black, blue, tur, grey) were identified, with kME values of genes calculated for each module (Fig. 8B). Under drought stress, the 276 genes specifically regulated by MT exhibited module-specific enrichment within the blue and turquoise module (Fig. 8C). Further analysis of the 276 genes identified 12 candidate genes exhibiting MT-induced expression under drought stress and a module membership (kME) >0.6. These included PIF8, MYC5, AUX1, ribonuclease H-like superfamily protein, NB-ARC domain-containing disease resistance protein, HMP20, RLK4, KCO1, GOX2, PMDH2, GIM2, and RCA (Table S5). These genes spanned functionally characterized categories, such as transcriptional regulation, disease resistance, and transport processes, underscoring their probable significance in MT-driven drought adaptation. Transcription factors can regulate the expression of numerous genes and play critical regulatory roles in various biological processes. Therefore, among the 12 core genes, we focused on two genes encoding BHLH transcription factors (GhPIF8 and GhMYC5). These two genes showed significant co-expression correlations with multiple genes (Figs. 9A and 9B), suggesting that they may act as hub genes in the MT-activated drought resistance network.

Figure 8 Construction and analysis of WGCNA.

(A) Venn diagram of DEGs comparing CK vs. DS, CK vs. CK_MT, DS vs. DS_MT. (B) Co-expression modules (clusters) identified through WGCNA. (C) Venn diagram illustrating the 276 DEGs specifically induced by MT in relation to the four identified modules.

Figure 9 Dentification of core genes in gene co-expression modules.

(A) GhPIF8 in the turquoise module, with lines representing the expression correlation between the hub gene and other genes. (B) GhMYC5 in the turquoise module, with lines representing the expression correlation between the hub gene and other genes.

Discussion

Escalating anthropogenic climate perturbations have positioned hydric deficit stress as a principal constraint on crop growth and development (Gupta, Rico-Medina & Caño-Delgado, 2020). Empirical investigations reveal hydric deficit induces redox homeostasis perturbation through reactive oxygen species (ROS) hyperaccumulation, resulting in plasma membrane lipid peroxidation and activation of multiple stress signaling pathways, while also causing leaf wilting, chlorosis, and growth inhibition in cotton, ultimately reducing yield and fiber quality (Mahmood et al., 2019; Zhu et al., 2024a). Utilizing plant growth regulators represents a principal agrobiotechnological intervention for drought stress amelioration (Godínez-Mendoza et al., 2023), and MT has been demonstrated to significantly reduce the detrimental impacts of drought on plants (Jahan et al., 2024). Exogenous MT supplementation (DS_MT) significantly enhanced the growth phenotype of cotton subjected to drought stress. Compared to the drought-stressed control group (DS), the DS_MT group exhibited higher leaf RWC, PRO content, and SOD, POD and CAT enzyme activities, along with a significant reduction in MDA content, suggesting that MT alleviates physiological damage caused by drought by regulating osmotic balance and the antioxidant system. Transcriptome analysis further revealed that drought-induced DEGs specifically enriched by MT were associated with biological processes such as organic acid metabolic process, response to abiotic stimulus, and amino acid metabolic process, as well as KEGG pathways including circadian rhythm—plant, thiamine metabolism and taurine and hypotaurine metabolism. These findings imply that MT enhances drought tolerance in cotton by modulating metabolic reprogramming and stress signaling responses. Furthermore, WGCNA analysis identified 12 hub genes from 276 MT-specifically regulated genes, among which two BHLH transcription factors (GhPIF8 and GhMYC5) showed significant co-expression correlations with multiple genes, highlighting their potential roles as key effectors in MT-mediated drought signal transduction (Figs. 8A and 8B).

MT enhances the antioxidant system of cotton under drought stress

Drought stress-mediated hyperaccumulation of ROS provokes oxidative damage to cellular architecture, thereby impeding plant morpho-physiological ontogeny (Mahmood et al., 2019). Previous evidence demonstrates that exogenous MT supplementation attenuates ROS accumulation and MDA via coordinated activation of enzymatic antioxidants, conferring enhanced oxidative stress resilience in plants (Li et al., 2019b; Lu et al., 2022; Zhang et al., 2023; Talaat, 2023; Chen et al., 2024). In this study, the DS_MT cohort exhibited 28.6%, 30.8% and 33% enhancement in leaf CAT, POD and SOD activities, respectively, relative to DS controls (Figs. 1D, 1E, 1F), indicating a mechanistic role for MT in facilitating redox equilibrium through enzymatic antioxidant machinery activation. Additionally, the MDA content in the DS_MT group decreased by 31.2% compared to the DS group (Fig. 1G), indicating that MT alleviates membrane lipid peroxidation damage in cotton under drought conditions. As a key osmotic regulator (Ozturk et al., 2021), PRO accumulation in the DS_MT group increased by 21.9% compared to the DS group (Fig. 1C), suggesting that MT supports osmotic balance in cells by enhancing the synthesis of osmotic adjustment substances. In summary, MT confers drought tolerance to cotton through synergistic potentiation of enzymatic ROS-scavenging systems, attenuation of membrane lipid peroxidative damage, and upregulation of osmoregulatory metabolite biosynthesis.

MT regulates multiple metabolic pathways to enhance cotton’s resistance to drought stress

Previous investigations have established that MT mitigates phytotoxic effects of environmental stress in plants through multifaceted regulation of stress-responsive metabolic networks (Tiwari et al., 2021; Dzinyela et al., 2024). Transcriptomic profiling reveals exogenous MT augments plant adaptive capacity to abiotic stress through orchestration of phenylpropanoid-derived flavonoid biosynthesis and phenolic compound metabolic flux (Jafari & Shahsavar, 2021; Duan et al., 2024). MT can enhance drought tolerance in wheat by regulating tryptophan metabolism and flavonoid biosynthesis, promoting the biosynthesis of jasmonic acid and lignin, and upregulating the expression of transcription factors HY5 and MYB86 (Luo et al., 2023a; Li et al., 2024; Shaffique et al., 2024). In rice, MT regulates drought response by activating antioxidant reactions, synergizing with transcription factors, and modulating plant hormone pathways (Li et al., 2022c; Mao et al., 2025). For maize, exogenous MT enhances its drought tolerance through multiple mechanisms. It activates transcription factor families, thereby regulating glutathione metabolism, calcium signal transduction, jasmonic acid biosynthesis, flavonoid synthesis pathways, and phytohormone signaling (Zhao et al., 2021b; Wang et al., 2023a). Additionally, it promotes stomatal opening, regulates carbon and nitrogen metabolism as well as the expression of related genes, and improves photosynthetic capacity, sucrose biosynthesis, and protein biosynthesis, ultimately enhancing drought tolerance (Zhao et al., 2021a; Ren et al., 2021). In this study, GO enrichment analysis revealed that under drought conditions, the DEGs induced by MT were specifically enriched in biological processes such as organic acid metabolic process, carboxylic acid metabolic process, small molecule biosynthetic process, response to abiotic stimulus, and amino acid metabolic process. KEGG pathway enrichment analysis indicated that the DEGs induced by MT were specifically were significantly enriched in three pathways: Circadian rhythm - plant, Thiamine metabolism, and Taurine and hypotaurine metabolism (Fig. 6C). Compared with crops such as maize, rice, and wheat, we identified novel metabolic pathways specifically regulated by exogenous MT: Circadian rhythm—plant, Thiamine metabolism, and Taurine and hypotaurine metabolism—which confer drought resistance in cotton.

Cotton’s broad—spectrum stress tolerance is markedly improved via the modulation of circadian rhythm, thiamine and galactose metabolism, as well as the biosynthesis of carotenoids, phenylpropanoids, flavonoids, and zeatin. Meanwhile, MAPK signaling pathway is activated (Li et al., 2022b). Furthermore, exogenous MT enhances cold stress resilience in Hulless barley through transcriptional reinstatement of circadian oscillator regulatory networks under low-temperature imposition (Chang et al., 2021). Concurrently, CIRCADIAN CLOCK ASSOCIATED1 (OsCCA1) mediates multistress resilience in rice under polyvalent abiotic stress conditions through molecular orchestration of ABA signal transduction dynamics (Wei et al., 2022b). Studies have found that CHS, PIF8, CRY1, LHY1, and HY5 are all highly associated with plant stress resistance (Gao et al., 2015; He et al., 2020; Yang et al., 2023; Ahmad et al., 2023; Lu et al., 2023; Liu et al., 2025b). Extensive research has confirmed that the circadian rhythm—plant pathway plays a key role in the stress response of organisms, and the genes encoding CHS, PIF8, CRY1, LHY1, and HY5 can effectively regulate plant stress resistance, indicating that these genes may play a crucial role in the drought response of cotton. Transcriptome analysis reveals that, under drought conditions, the DEGs regulated by MT are specifically enriched in the “circadian rhythm—plant” pathway (Fig. 6C). This suggests that MT may enhance cotton’s tolerance to drought stress by maintaining its circadian rhythm. The results of this study further validate that this pathway exerts a core regulatory function in organisms’ coping with stress.

Under drought stress conditions, the DEGs induced by MT were significantly enriched in the thiamine metabolism pathway (Figs. 6C and 7A). Studies have demonstrated that thiamine functions not only as an enzymatic cofactor in central metabolic processes including glycolysis, tricarboxylic acid cycle, and pentose phosphate pathway (Goyer, 2010), but also participates in mediating stress adaptation mechanisms in plants (Fitzpatrick & Chapman, 2020), establishing its indispensable role in maintaining plant physiological functions. Exogenous application of thiamine has been shown to mitigate the detrimental effects of drought on Pterocarya stenoptera, stimulating chlorophyll synthesis and the expression of photosynthesis-related genes (Zhang, Wang & Li, 2023). In maize, total thiamine content in plant tissues significantly increases following stress application (Rapala-Kozik, Kowalska & Ostrowska, 2008). Experimental evidence reveals that foliar supplementation of thiamine stimulates maize growth, augments the activity of antioxidant enzymes, and elevates photosynthetic pigment concentrations, thereby effectively mitigating abiotic stress-induced physiological impairments (Kaya et al., 2014). Additionally, thiamine improves drought tolerance in peas by augmenting various biochemical traits, including antioxidant enzyme activities, chlorophyll content, secondary metabolites, polyamines, and nutrient levels. In studies involving Vicia faba, thiamine potentiates salinity tolerance through modulating intracellular antioxidant and osmoprotectant biosynthesis (Ahmed & Sattar, 2024). Oil palm responds to oxidative stress by upregulating the expression of genes THIC and THI1/THI4, which are involved in thiamine biosynthesis (Idris et al., 2018). MsTHI1 can enhance drought tolerance in alfalfa (Medicago sativa L.) through mechanisms such as improving photosynthesis, regulating the antioxidant defense system, maintaining osmotic homeostasis, and mediating plant signal transduction (Yin et al., 2022). Overexpression of the DXS gene enhances stress resistance in poplar (Wei et al., 2019). Existing studies have demonstrated that the thiamine metabolism pathway is a key pathway for organisms to cope with stress, and the genes encoding Thi1 and DXS can effectively improve plant stress resistance, suggesting that these genes may play a crucial role in the drought response of cotton. Our findings indicate that under drought stress, the expression levels of the enzymes Thi1, DXS, and ThiC, which are crucial for thiamine synthesis, are significantly upregulated. Consistent with previous research results, experimental data establish that MT potentiates drought tolerance in cotton through regulation of thiamine biosynthesis pathways.

Taurine is essential for maintaining the normal function of the electron transport chain, preserving glutathione levels, activating antioxidant responses, and enhancing membrane stability (Baliou et al., 2021). This molecule serves as a central regulator of plant abiotic stress tolerance mechanisms (Ashraf et al., 2023b). Baseggio Conrado et al. (2021) further demonstrated that hypotaurine exerts protective effects against cellular damage induced by ultraviolet A (UVA) radiation. Exogenous application of taurine modulates oxidative defense kinetics, secondary metabolic pathways, and nutrient homeostasis (Ashraf et al., 2022; Ashraf et al., 2023a). Furthermore, biochemical analyses confirm that taurine augments potassium (K), phosphorus (P), and calcium (Ca2+) acquisition, consequently potentiating heavy metal stress tolerance in plants (Hafeez et al., 2022). Physiological investigations in Pisum sativum demonstrate that taurine enhances osmotic homeostasis and antioxidant defense systems, thereby reducing cellular dehydration and lipid peroxidation under stress exposure while counteracting salt toxicity and iron-deprivation pathologies (Ashraf et al., 2023b). Proteomic studies have demonstrated that salt-tolerant oat varieties enhance their salt tolerance by regulating the metabolism of taurine and hypotaurine (Chen et al., 2022). A study by Wang et al. (2023b) revealed that FMO1 regulates ROS homeostasis through direct interaction with SlCAT2 and exerts a negative regulatory effect on drought tolerance in tomato via an ABA-dependent pathway. Previous studies have demonstrated that the taurine and hypotaurine metabolism pathway is associated with stress responses, and among these, genes encoding FMO can effectively improve plant stress resistance, suggesting that such genes may play a crucial role in the drought response of cotton. Experimental findings demonstrate that MT suppresses three FMO and activates two GGT, establishing a mechanistic link between MT-mediated drought stress tolerance and taurine and hypotaurine metabolism in cotton systems. This result further confirms that the taurine and hypotaurine metabolism pathway is a key pathway for organisms to cope with stress.

Numerous studies have established that the pathways associated with circadian rhythm—plant, thiamine metabolism, and taurine and hypotaurine metabolism are essential for mediating plant responses to abiotic stress. Transcriptome analyses confirm that MT mechanistically potentiates drought stress resilience in cotton via targeting the modulation of these pathways, thereby establishing a functional hierarchy in stress-adaptive pathway modulation. Collectively, these findings provide compelling evidence that circadian rhythm—plant, thiamine metabolism, and taurine and hypotaurine metabolism are key mechanisms by which plants cope with abiotic stress.

Li et al. (2019a) reported that MT enhances cotton resistance to Verticillium dahliae primarily by regulating genes involved in lignin and gossypol biosynthesis within the phenylpropanoid, MVA, and gossypol pathways. Foliar application of MT can enhance drought tolerance during the boll stage of cotton by regulating sugar metabolism and nitrogen metabolism pathways (Khattak et al., 2023). In contrast, our study revealed that MT improves drought tolerance in cotton seedlings by regulating the circadian rhythm, thiamine metabolism, and taurine and hypotaurine metabolism pathways. This study, together with previous research, demonstrates that MT enhances cotton’s tolerance to both biotic and abiotic stresses by targeting distinct molecular pathways. Notably, MT also exhibits developmental stage-dependent activation of diverse signaling networks, which collectively improve drought resilience across critical growth phases (e.g., seedling establishment, flowering, and boll development).

MT modulates the expression of pivotal genes in response to drought stress

Deciphering the hub genes governing MT-mediated drought adaptation in Gossypium species reveals unprecedented stress resilience networks, thereby identifying biotechnological targets for precision-bred climate-resilient cultivars through molecular dissection of osmoregulatory signaling cascades. For example, in loquat plants, MT treatment promotes ABA-mediated stomatal closure by upregulating EjWRKY17 expression (Wang et al., 2021). Additionally, in cold-stored wolfberries, MT activates the LbaGATA transcription factor, which regulates the synthesis of flavonoids and carotenoids (Jiang et al., 2025). In apple fruit, MT delays ethylene production by inhibiting MdREM10 expression (Li et al., 2025c). Previous studies have shown that MT can play a critical regulatory role in various biological processes by modulating the expression of downstream genes. Transcriptomic profiling identified 276 genes with significant differential expression profiles in the DS_MT cohort versus DS controls through comparative analysis of MT-mediated stress adaptation networks (Fig. 8A). To delineate central regulatory nodes within MT-mediated drought adaptation networks, we implemented WGCNA to establish drought-responsive interactomes, followed by eigengene connectivity (kME) topology analyses for hub gene identification. The results of the co-expression analysis indicated that the DEGs could be categorized into four distinct expression modules (Fig. 8B). Module-driven analysis revealed that 276 MT-modulated genes under drought exposure predominantly localized to the turquoise co-expression network, exhibiting the highest eigengene connectivity (Fig. 8C). Based on the connectivity values of the kME > 0.6, we identified two bHLH transcription factors (GhPIF8 and GhMYC5) that were significantly induced by MT. These factors exhibited prominent core node characteristics within the module (Figs. 9A and 9B), suggesting their potential role as key regulatory elements in the response to drought stress mediated by MT. Molecular genetic evidence demonstrates that CsPIF8 transcriptionally activates CsSOD, potentiating low-temperature resilience in Citrus through enhanced superoxide dismutase-mediated redox homeostasis (He et al., 2020). Furthermore, PtbHLH2 8 (a homolog of MYC5) activates transcription by binding to the promoter of PtCOMT5, thereby modulating MT synthesis and root development, ultimately improving drought tolerance in citrus (Zhu et al., 2025). These studies indicate that the bHLH transcription factors PIF8 and MYC5 can significantly enhance plant stress tolerance. The present study reveals that GhPIF8 and GhMYC may be key responsive genes involved in MT-mediated improvement of drought resistance in cotton, which further indicates that these two genes play important roles in plant drought responses. Future in-depth functional studies are required to verify the function of GhPIF8 and GhMYC5 as core transcriptional regulators in MT-dependent drought resistance signaling in cotton.

This study demonstrates that exogenous MT can enhance drought resistance in cotton by increasing antioxidant enzyme activity, elevating PRO content, reducing MDA accumulation, regulating the circadian rhythm—plant, thiamine metabolism, and taurine and hypotaurine metabolism pathways, and upregulating the expression of transcription factors GhPIF8 and GhMYC5 (Fig. 10). This provides a theoretical reference for the future application of MT in cotton cultivation.

Figure 10 Mechanism by which MT enhances drought resistance in cotton.

The study found that exogenous foliar application of MT can improve drought resistance in cotton seedlings, suggesting that this spraying method might be used to cope with drought stress in cotton in the future. However, the current high cost of MT means that its practical application in agricultural production still faces challenges.

Conclusions

In this study, we establishes the mechanistic basis underlying exogenous MT-mediated drought resilience in cotton, integrating physiological adaptation dynamics with molecular determinants of hydric stress mitigation. The phenotypic and physiological results demonstrated that MT significantly enhanced the activities of antioxidant enzymes, specifically SOD, POD and CAT, reduced MDA accumulation, and improved osmotic adjustment capacity mediated by PRO, effectively alleviating drought-induced damage to cotton. Transcriptome analysis revealed that MT specifically activated the expression of genes associated with circadian rhythm—plant, thiamine metabolism, and taurine and hypotaurine metabolism under drought conditions. Additionally, a comparison between the DS_MT and DS groups identified 276 genes that specifically responded to MT. Co-expression network analysis further identified two transcription factors from the bHLH family, GhPIF8 and GhMYC5, suggesting their potential role as core response genes mediating the enhancement of drought resistance in cotton induced by MT. This study reveals the effect of MT on drought resistance in cotton and its potential regulatory mechanisms, establishing a theoretical framework for MT-mediated strategies to alleviate drought stress in cotton production.

In the future, it will be necessary to explore the effects of MT on other abiotic stresses (such as heat stress, cold stress, flood stress, salt stress, and alkali stress) and their potential regulatory mechanisms. This will provide a theoretical reference for mitigating damage to cotton caused by abiotic stresses through the application of MT in agricultural production.

Supplemental Information

Supplemental Information 1 PCA and Heatmap Analysis of Transcriptomic Gene Expression Patterns

Supplemental Information 2 Heatmap of expression levels of DEGs encoding SOD, POD, and CAT in DS_MT vs DS

Supplemental Information 3 Relative expression levels of 10 genes examined by qRT-PCR. Data are means ±SD of n = 3 independent experiments

Supplemental Information 4 qRT-PCR primers for 10 DEGs specifically induced by MT under drought stress

Supplemental Information 5 RNA Quality of Cotton (Gossypium hirsutum)

Supplemental Information 6 Transcriptome Sequencing Data Quality Assessment and GC Content Analysis

Supplemental Information 7 12 Genes Specifically Induced by MT Under Drought Stress and Their Functional Annotations

Supplemental Information 8 Transcriptome data of 10 DEGs specifically induced by MT under drought stress

Supplemental Information 9 Raw data

Throughout the writing of this dissertation, I have received a great deal of support and assistance. First and foremost, I would like to thank the anonymous reviewers for their insightful and helpful remarks.

Additional Information and Declarations

Competing Interests

Author Contributions

DNA Deposition

Data Availability

The authors declare there are no competing interests.

Xingyue Zhong performed the experiments, analyzed the data, prepared figures and/or tables, authored or reviewed drafts of the article, and approved the final draft.

Aixia Han performed the experiments, analyzed the data, prepared figures and/or tables, authored or reviewed drafts of the article, and approved the final draft.

Yunhao Liusui performed the experiments, prepared figures and/or tables, and approved the final draft.

Xin Zhang performed the experiments, prepared figures and/or tables, and approved the final draft.

Wanwan Fu performed the experiments, prepared figures and/or tables, and approved the final draft.

Ziyu Wang performed the experiments, prepared figures and/or tables, and approved the final draft.

Yuanxin Li performed the experiments, prepared figures and/or tables, and approved the final draft.

Jing Cao performed the experiments, prepared figures and/or tables, and approved the final draft.

Yanjun Guo conceived and designed the experiments, analyzed the data, prepared figures and/or tables, authored or reviewed drafts of the article, and approved the final draft.

JingBo Zhang conceived and designed the experiments, analyzed the data, prepared figures and/or tables, authored or reviewed drafts of the article, and approved the final draft.

The following information was supplied regarding the deposition of DNA sequences:

The raw sequence data reported in this paper are available at the Genome Sequence Archive (Genomics, Proteomics & Bioinformatics 2021) in National Genomics Data Center (Nucleic Acids Res 2022), China National Center for Bioinformation/Beijing Institute of Genomics, Chinese Academy of Sciences, GSA: CRA025657.

The following information was supplied regarding data availability:

The raw datasets supporting all figures and results are available in the Supplementary File.

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
