# Peer review of "Identification of key pathways and genes underlying melatonin-enhanced drought tolerance in cotton"

_PeerJ, doi:10.7717/peerj.20005_

## Round 0.1 · original submission · Major Revisions

All reviewers raise concerns about experimental design and statistical and bioinformatic analysis. While it is not mandatory to follow their suggestions, the authors' choices and decisions should be justified.

Reviewer 1 ·

Basic reporting

This study provides valuable insights into melatonin (MT)-mediated drought tolerance mechanisms in cotton through integrated physiological and transcriptomic analyses. The identification of key metabolic pathways (circadian rhythm, thiamine metabolism, taurine/hypotaurine metabolism) and transcription factors (GhPIF8, GhMYC5) represents a significant contribution to stress physiology. However, major revisions are required to address methodological gaps, statistical inconsistencies, and contextual framing. The work has strong potential but currently falls short of PeerJ's standards for experimental rigor and clarity. Major issues and revisions are listed below:
.

Experimental design

1. Experimental design & methodology
- No data on soil water content, leaf water potential, or PEG concentration used for drought simulation. Include metrics (e.g., % soil moisture, Ψleaf) to define drought severity.
- RNA-seq used 3 biological replicates, but physiological assays state "triplicate measurements" without clarifying if these were technical/biological. Clarify replicate types (biological vs. technical) in Methods; provide n-values for all assays in figure legends.
- Discontinuous foliar application (days 1,3,5-8) lacks justification. Potential for wash-off during drought not addressed. Justify the interval design; include control for MT persistence (e.g., quantify MT in leaves post-application).

2. Data analysis & statistics
- Student’s t-test used for multi-group comparisons (Fig. 1). Replace with ANOVA + Tukey’s HSD.
- The 276 MT-specific DEGs were filtered to 12 hub genes (kME > 0.6), but only GhPIF8/GhMYC5 were discussed. Include a full list of 12 genes in a table with functional annotations; justify focus on bHLH factors.
- No qRT-PCR validation of DEGs (e.g., Thi4, FMO1). Validate key DEGs via qRT-PCR.

Validity of the findings

3. Results interpretation
- Taurine metabolism’s role in plants is weakly supported (cited animal studies: Baliou et al. 2021). Cite plant-specific literature or temper claims.
- SOD/POD activities increased with MT (Fig. 1), but no linked DEGs (e.g., SOD genes) were highlighted. Integrate enzyme data with relevant DEGs (e.g., via correlation analysis).
- Fig. 1 is described as having a "heatmap" in text (p. 11), but only bar graphs are shown. Fig. 7 heatmap labels (Thi4, DXS) do not match gene IDs in the text. Correct figure annotations and align with text.

4. Contextual & literature gaps
- Limited discussion of prior work on MT in cotton (e.g., Li et al. 2019a on Verticillium resistance). Contrast findings with existing MT mechanisms (e.g., phenylpropanoid vs. thiamine pathways).
- Are circadian/thiamine/taurine pathways uniquely regulated by MT or general drought responses? Compare DS vs. DS_MT KEGG results to CK vs. DS to highlight MT-specificity.
- Cite some more references to improve the context and discussion: e.g., Field Crops Research 261 (2021) 107989; 2024, 306:109217.

Additional comments

5. Data accessibility
- RNA-seq data is deposited (CRA025657), but physiological raw data and image source files are missing. Upload all raw data (physiological measurements, unprocessed images) to a public repository

Reviewer 2 ·

Basic reporting

-

Experimental design

-

Validity of the findings

-

Additional comments

In the manuscript, the authors investigated the Identiûcation of key pathways and genes underlying melatonin-enhanced drought tolerance in cotton, and their findings are very interesting and valuable for future research on melatonin. The way of presentation is fairly good. However, I have several comments to improve the manuscript-

1. The lines 18-21 and 35-36 are not clear, and the authors should shortly include the experimental treatments in the abstract.

2. The keywords should not be like those in the title.

3. There are many irrelevant sentences in the introduction, and there is a lack of coherence among the sentences. In-depth and insightful addressing is missing in the introduction. The introduction should be refined with recent relevant references such as doi: 10.1093/plphys/kiad317, doi: 10.1093/plphys/kiad027

4. Introduction section meed to rigorous modification.

5. Is this cotton cultivar drought tolerant or sensitive?

6. How many day-old seedlings were exposed to drought stress?

7. Subheading - Determination of Physiological Indices of Cotton Plants under Drought Stress- should change; the methodology of this heading must be elaborately explained. This subheading should be split and give new headings, including statistical analysis.

8. Figure 1A: There is no significant variation between the DS and DS_MT.

9. -The discussion is fairly well explained and verified with previous research, but it needs further improvement with updated references and coherence among the studied findings.

10. It would be better to draw a model figure to highlight the mechanism.

11. There are a lot of typos and grammar errors throughout the manuscript. The authors should go over the entire MS carefully to ensure proper use of English.

Reviewer 3 ·

Basic reporting

-

Experimental design

-

Validity of the findings

-

Additional comments

The authors have investigated the role of exogenous melatonin in enhancing drought tolerance in cotton (Gossypium hirsutum L.) by exploring both physiological and molecular mechanisms. Through integrated physiological profiling and high-throughput transcriptomic sequencing, the study investigated key pathways and genes regulated by melatonin under drought stress. Their findings suggest that melatonin improves cotton's drought resilience by modulating antioxidant systems, osmotic regulation, and specific metabolic pathways.

The manuscript requires major revision for consideration for publication in this journal. The following are my comments for revision of the manuscript to ensure the clarity and completeness, as follows:

1. Introduction
In the Introduction, it is necessary to explicitly state the gaps in the existing literature regarding melatonin's role in drought tolerance in cotton. Providing additional justification for why this study is necessary would enhance the clarity. There are recent articles addressing melatonin’s role in regulating drought tolerance in cotton, and the authors should appropriately cite these studies to position their work as an update in the field.

2. Experimental Design
The study uses a 2 × 2 factorial design. Was this design the most appropriate for testing the hypothesis? Could the authors provide reasoning for choosing this design over others?

Why was the phyllosphere application method chosen for melatonin treatment? Is this method the most effective for ensuring that melatonin reaches the plant tissues involved in drought stress responses, compared to other possible methods like soil application or root absorption?

How was uniformity in the application of melatonin on the leaves ensured, and do you foresee any challenges in achieving this uniformity?

The experimental methods have to be explained more elaborately.

3. Data Analysis and Statistical Methods
The WGCNA analysis identified key regulatory modules, but how were the thresholds for module membership (kME) set? Was this based on previous studies, or was it an arbitrary selection? Could the authors justify why the kME > 0.6 was used?

The author should provide the RIN score, if done. If not done, how was RNA integrity measured?
Were the biological replicates for RNA-seq adequately distributed across the different treatments (CK, DS, DS_MT, and CK_MT)? How was technical variability minimized during RNA extraction and sequencing?

4. Functional Analysis and Pathways
The KEGG pathway analysis revealed pathways associated with plant circadian rhythm, thiamine metabolism, and taurine metabolism. Could the authors explain why these particular pathways were highlighted and their specific relevance to drought tolerance in cotton?

In the GO enrichment analysis, could the authors provide more insight into which specific genes within the Circadian rhythm - plant, Thiamine metabolism, and Taurine and hypotaurine metabolism pathways are most critical for drought response in cotton?

How do the findings regarding bHLH transcription factors (GhPIF8 and GhMYC5) compare to similar findings in other species? Is there any existing literature suggesting these transcription factors' roles in drought tolerance or melatonin-mediated stress responses?

5. Discussion and Application
The Discussion provides valuable insights into the mechanisms behind MT’s effects on cotton under drought stress. However, could the authors provide more specific examples of how melatonin could be applied in cotton cultivation practices? Are there any limitations or challenges that farmers may face in implementing MT treatment in a small paragraph as a future goal?

How does this study compare to previous research on melatonin’s role in enhancing drought tolerance in other crops? Could the authors provide a more in-depth comparison of their findings with those from similar studies in plants like maize, rice, or wheat?

6. Conclusion and Future Directions
Could the Conclusions section be expanded to summarize not just the findings but also future research directions? What are the next steps in investigating melatonin’s role in other stress tolerance mechanisms in cotton or other crops?

---

## Round 0.2 · accepted · Accept

The reviewers are satisfied with the revised manuscript and have no further comments.

Reviewer 1 ·

Basic reporting

The authors have addressed the comments properly and revised the manuscript accordingly. I have no further comments.

Experimental design

It seems to meet the requirements.

Validity of the findings

It is good enough

Additional comments

no

Reviewer 2 ·

Basic reporting

no comment

Experimental design

no comment

Validity of the findings

no comment

Additional comments

The authors have thoroughly addressed all reviewer comments and revised the manuscript accordingly. Please pay attention in your replies to comments in the future, for every comment in your "ANSWER or RESPONSE", please include the line numbers of the changes in the text. This allows the Reviewers to quickly see where you made changes in the text for a given Reviewer Comment. I recommend Accept.